# GraphCodeBERT: Pre-training Code Representations with Data Flow

**Daya Guo**[1]*, **Shuo Ren**[2]*, **Shuai Lu**[3]*, **Zhangyin Feng**[4]*, **Duyu Tang**[5], **Shujie Liu**[5], **Long Zhou**[5], **Nan Duan**[5], **Alexey Svyatkovskiy**[6], **Shengyu Fu**[6], **Michele Tufano**[6], **Shao Kun Deng**[6], **Colin Clement**[6], **Dawn Drain**[6], **Neel Sundaresan**[6], **Jian Yin**[1], **Daxin Jiang**[7], and **Ming Zhou**[5]

[1]School of Computer Science and Engineering, Sun Yat-sen University.
[2]Beihang University, [3]Peking University, [4]Harbin Institute of Technology,
[5]Microsoft Research Asia, [6]Microsoft Devdiv, [7]Microsoft STCA

## Abstract

Pre-trained models for programming language have achieved dramatic empirical improvements on a variety of code-related tasks such as code search, code completion, code summarization, etc. However, existing pre-trained models regard a code snippet as a sequence of tokens, while ignoring the inherent structure of code, which provides crucial code semantics and would enhance the code understanding process. We present GraphCodeBERT, a pre-trained model for programming language that considers the inherent structure of code. Instead of taking syntactic-level structure of code like abstract syntax tree (AST), we use data flow in the pre-training stage, which is a semantic-level structure of code that encodes the relation of "where-the-value-comes-from" between variables. Such a semantic-level structure is less complex and does not bring an unnecessarily deep hierarchy of AST, the property of which makes the model more efficient. We develop GraphCodeBERT based on Transformer. In addition to using the task of masked language modeling, we introduce two structure-aware pre-training tasks. One is to predict code structure edges, and the other is to align representations between source code and code structure. We implement the model in an efficient way with a graph-guided masked attention function to incorporate the code structure. We evaluate our model on four tasks, including code search, clone detection, code translation, and code refinement. Results show that code structure and newly introduced pre-training tasks can improve GraphCodeBERT and achieves state-of-the-art performance on the four downstream tasks. We further show that the model prefers structure-level attentions over token-level attentions in the task of code search.[1]

## 1 Introduction

Pre-trained models such as ELMo (Peters et al., 2018), GPT (Radford et al., 2018) and BERT (Devlin et al., 2018) have led to strong improvement on numerous natural language processing (NLP) tasks. These pre-trained models are first pre-trained on a large unsupervised text corpus, and then fine-tuned on downstream tasks. The success of pre-trained models in NLP also promotes the development of pre-trained models for programming language. Existing works (Kanade et al., 2019; Karampatsis & Sutton, 2020; Feng et al., 2020; Svyatkovskiy et al., 2020; Buratti et al., 2020) regard a source code as a sequence of tokens and pre-train models on source code to support code-related tasks such as code search, code completion, code summarization, etc. However, previous works only utilize source code for pre-training, while ignoring the inherent structure of code. Such code structure provides useful semantic information of code, which would benefit the code understanding process. Taking the expression $v = max\_value - min\_value$ as an example, $v$ is computed from $max\_value$ and $min\_value$. Programmers do not always follow the naming conventions so that it's hard to understand the semantic of the variable $v$ only from its name. The semantic structure of code provides a way to understand the semantic of the variable $v$ by leveraging dependency relation between variables.

---

*Work done while this author was an intern at Microsoft Research Asia. Contact: Daya Guo (guody5@mail2.sysu.edu.cn)

[1]All the codes and data are available at `https://github.com/microsoft/CodeBERT`.

In this work, we present GraphCodeBERT, a pre-trained model for programming language that considers the inherent structure of code. Instead of taking syntactic-level structure of code like abstract syntax tree (AST), we leverage semantic-level information of code, i.e. data flow, for pre-training. Data flow is a graph, in which nodes represent variables and edges represent the relation of "where-the-value-comes-from" between variables. Compared with AST, data flow is less complex and does not bring an unnecessarily deep hierarchy, the property of which makes the model more efficient. In order to learn code representation from source code and code structure, we introduce two new structure-aware pre-training tasks. One is data flow edges prediction for learning representation from code structure, and the other is variable-alignment across source code and data flow for aligning representation between source code and code structure. GraphCodeBERT is based on Transformer neural architecture (Vaswani et al., 2017) and we extend it by introducing a graph-guided masked attention function to incorporate the code structure.

We pre-train GraphCodeBERT on the CodeSearchNet dataset (Husain et al., 2019), which includes 2.3M functions of six programming languages paired with natural language documents. We evaluate the model on four downstream tasks: natural language code search, clone detection, code translation, and code refinement. Experiments show that our model achieves state-of-the-art performance on the four tasks. Further analysis shows that code structure and newly introduced pre-training tasks can improve GraphCodeBERT and the model has consistent preference for attending data flow.

In summary, the contributions of this paper are: (1) GraphCodeBERT is the first pre-trained model that leverages semantic structure of code to learn code representation. (2) We introduce two new structure-aware pre-training tasks for learning representation from source code and data flow. (3) GraphCodeBERT provides significant improvement on four downstream tasks, i.e. code search, clone detection, code translation, and code refinement.

## 2 RELATED WORKS

**Pre-Trained Models for Programming Languages**  Inspired by the big success of pre-training in NLP (Devlin et al., 2018; Yang et al., 2019; Liu et al., 2019; Raffel et al., 2019), pre-trained models for programming languages also promotes the development of code intelligence (Kanade et al., 2019; Feng et al., 2020; Karampatsis & Sutton, 2020; Svyatkovskiy et al., 2020; Buratti et al., 2020). Kanade et al. (2019) pre-train a BERT model on a massive corpus of Python source codes by masked language modeling and next sentence prediction objectives. Feng et al. (2020) propose CodeBERT, a bimodal pre-trained model for programming and natural languages by masked language modeling and replaced token detection to support text-code tasks such as code search. Karampatsis & Sutton (2020) pre-train contextual embeddings on a JavaScript corpus using the ELMo framework for program repair task. Svyatkovskiy et al. (2020) propose GPT-C, which is a variant of the GPT-2 trained from scratch on source code data to support generative tasks like code completion. Buratti et al. (2020) present C-BERT, a transformer-based language model pre-trained on a collection of repositories written in C language, and achieve high accuracy in the abstract syntax tree (AST) tagging task.

Different with previous works, GraphCodeBERT is the first pre-trained model that leverages code structure to learn code representation to improve code understanding. We further introduce a graph-guided masked attention function to incorporate the code structure into Transformer and two new structure-aware pre-training tasks to learn representation from source code and code structure.

**Neural Networks with Code Structure**  In recent years, some neural networks leveraging code structure such as AST have been proposed and achieved strong performance in code-related tasks like code completion (Li et al., 2017; Alon et al., 2019; Kim et al., 2020), code generation (Rabinovich et al., 2017; Yin & Neubig, 2017; Brockschmidt et al., 2018), code clone detection (Wei & Li, 2017; Zhang et al., 2019; Wang et al., 2020), code summarization (Alon et al., 2018; Hu et al., 2018) and so on (Nguyen & Nguyen, 2015; Allamanis et al., 2018; Hellendoorn et al., 2019). Nguyen & Nguyen (2015) propose an AST-based language model to support the detection and suggestion of a syntactic template at the current editing location. Allamanis et al. (2018) use graphs to represent programs and graph neural network to reason over program structures. Hellendoorn et al. (2019) propose two different architectures using a gated graph neural network and Transformers for combining local and global information to leverage richly structured representations of source code. However, these

works leverage code structure to learn models on specific tasks from scratch without using pre-trained models. In this work, we study how to leverage code structure for pre-training code representation.

## 3 DATA FLOW

In this section, we describe the basic concept and extraction of data flow. In next section, we will describe how to use data flow for pre-training.

Data flow is a graph that represents dependency relation between variables, in which nodes represent variables and edges represent where the value of each variable comes from. Unlike AST, data flow is same under different abstract grammars for the same source code. Such code structure provides crucial code semantic information for code understanding. Taking $v = max\_value - min\_value$ as an example, programmers do not always follow the naming conventions so that it is hard to understand the semantic of the variable. Data flow provides a way to understand the semantic of the variable $v$ to some extent, i.e. the value of $v$ comes from $max\_value$ and $min\_value$ in data flow. Besides, data flow supports the model to consider long-range dependencies induced by using the same variable or function in distant locations. Taking Figure 1 as an example, there are four variables with same name (i.e. $x^3$, $x^7$, $x^9$ and $x^{11}$) but with different semantic. The graph in the figure shows dependency relation between these variables and supports $x^{11}$ to pay more attention to $x^7$ and $x^9$ instead of $x^3$. Next, we describe how to extract data flow from a source code.

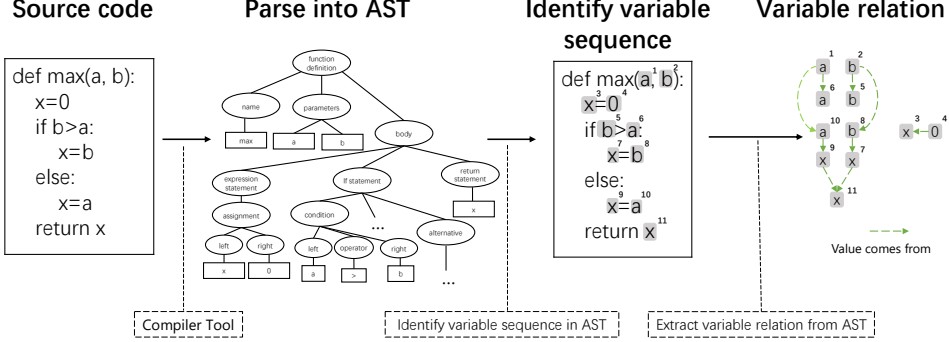

Figure 1: The procedure of extracting data flow given a source code. The graph in the rightmost is data flow that represents the relation of "where-the-value-comes-from" between variables.

Figure 1 shows the extraction of data flow through a source code. Given a source code $C = \{c_1, c_2, ..., c_n\}$, we first parse the code into an abstract syntax tree (AST) by a standard compiler tool[2]. The AST includes syntax information of the code and terminals (leaves) are used to identify the variable sequence, denoted as $V = \{v_1, v_2, ..., v_k\}$. We take each variable as a node of the graph and an direct edge $\varepsilon = \langle v_i, v_j \rangle$ from $v_i$ to $v_j$ refers that the value of $j$-th variable comes from $i$-th variable. Taking $x = expr$ as an example, edges from all variables in $expr$ to $x$ are added into the graph. We denote the set of directed edges as $E = \{\varepsilon_1, \varepsilon_2, ..., \varepsilon_l\}$ and the graph $\mathcal{G}(C) = (V, E)$ is data flow used to represent dependency relation between variables of the source code $C$.

## 4 GRAPHCODEBERT

In this section, we describe GraphCodeBERT, a graph-based pre-trained model based on Transformer for programming language. We introduce model architecture, graph-guided masked attention and pre-training tasks including standard masked language model and newly introduced ones. More details about model pre-training setting are provided in the Appendix A.

---

[2]https://github.com/tree-sitter/tree-sitter

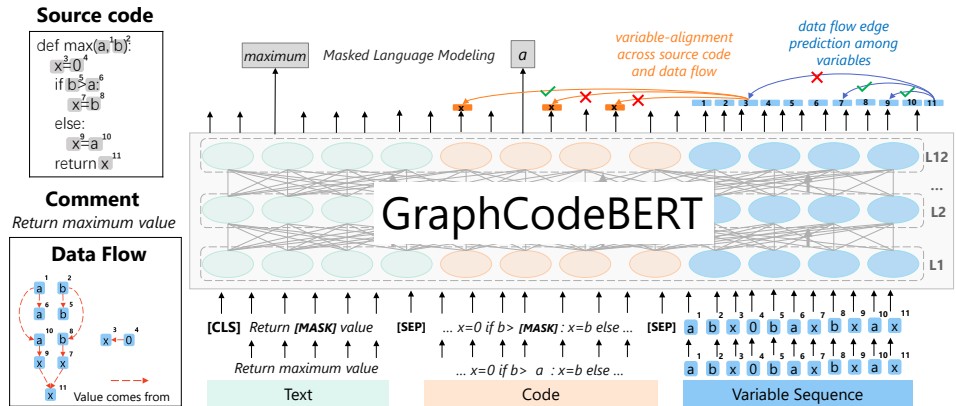

Figure 2: An illustration about GraphCodeBERT pre-training. The model takes source code paired with comment and the corresponding data flow as the input, and is pre-trained using standard masked language modeling (Devlin et al., 2018) and two structure-aware tasks. One structure-aware task is to predict where a variable is identified from (marked with orange lines) and the other is data flow edges prediction between variables (marked with blue lines).

## 4.1 MODEL ARCHITECTURE

Figure 2 shows the model architecture of GraphCodeBERT. We follow BERT (Devlin et al., 2018) and use the multi-layer bidirectional Transformer (Vaswani et al., 2017) as the model backbone. Instead of only using source code, we also utilize paired comments to pre-train the model to support more code-related tasks involving natural language such as natural language code search (Feng et al., 2020). We further take data flow, which is a graph, as a part of the input to the model.

Given a source code $C = \{c_1, c_2, ..., c_n\}$ with its comment $W = \{w_1, w_2, ..., w_m\}$, we can obtain the corresponding data flow $\mathcal{G}(C) = (V, E)$ as discussed in the Section 3, where $V = \{v_1, v_2, ..., v_k\}$ is a set of variables and $E = \{\varepsilon_1, \varepsilon_2, ..., \varepsilon_l\}$ is a set of direct edges that represent where the value of each variable comes from. We concatenate the comment, source code and the set of variables as the sequence input $X = \{[CLS], W, [SEP], C, [SEP], V\}$, where $[CLS]$ is a special token in front of three segments and $[SEP]$ is a special symbol to split two kinds of data types.

GraphCodeBERT takes the sequence $X$ as the input and then converts the sequence into input vectors $H^0$. For each token, its input vector is constructed by summing the corresponding token and position embeddings. We use a special position embedding for all variables to indicate that they are nodes of data flow. The model applies N transformer layers over the input vectors to produce contextual representations $H^n = transformer_n(H^{n-1}), n \in [1, N]$. Each transformer layer contains an architecturally identical transformer that applies a multi-headed self-attention operation (Vaswani et al., 2017) followed by a feed forward layer over the input $H^{n-1}$ in the $n$-th layer.

$$G^n = LN(MultiAttn(H^{n-1}) + H^{n-1}) \tag{1}$$
$$H^n = LN(FFN(G^n) + G^n) \tag{2}$$

where $MultiAttn$ is a multi-headed self-attention mechanism, $FFN$ is a two layers feed forward network, and $LN$ represents a layer normalization operation. For the $n$-th transformer layer, the output $\hat{G}^n$ of a multi-headed self-attention is computed via:

$$Q_i = H^{n-1}W_i^Q, \ K_i = H^{n-1}W_i^K, \ V_i = H^{n-1}W_i^V \tag{3}$$

$$head_i = \text{softmax}(\frac{Q_i K_i^T}{\sqrt{d_k}} + M)V_i \tag{4}$$

$$\hat{G}^n = [head_1; ...; head_u]W_n^O \tag{5}$$

where the previous layer's output $H^{n-1} \in \mathbb{R}^{|X| \times d_h}$ is linearly projected to a triplet of queries, keys and values using model parameters $W_i^Q, W_i^K, W_i^V \in \mathbb{R}^{d_h \times d_k}$, respectively. $u$ is the number of heads, $d_k$ is the dimension of a head, and $W_n^O \in \mathbb{R}^{d_h \times d_h}$ is the model parameters. $M \in \mathbb{R}^{|X| \times |X|}$ is a mask matrix, where $M_{ij}$ is 0 if $i$-th token is allowed to attend $j$-th token otherwise $-\infty$.

## 4.2 GRAPH-GUIDED MASKED ATTENTION

To incorporate the graph structure into Transformer, we define a graph-guided masked attention function to filter out irrelevant signals. The attention masking function could avoid the key $k_i$ attended by the query $q_j$ by adding the attention score $q_j^T k_i$ an infinitely negative value so that the attention weight becomes zero after using a softmax function. To represent dependency relation between variables, a node-query $q_{v_i}$ is allowed to attend to a node-key $k_{v_j}$ if there is a direct edge from the node $v_j$ to the node $v_i$ (i.e. $\langle v_j, v_i \rangle \in E$) or they are the same node (i.e. $i = j$). Otherwise, the attention is masked by adding an infinitely negative value into the attention score. To represent the relation between source code tokens and nodes of the data flow, we first define a set $E'$, where $\langle v_i, c_j \rangle / \langle c_j, v_i \rangle \in E'$ if the variable $v_i$ is identified from the source code token $c_j$. We then allow the node $q_{v_i}$ and code $k_{c_j}$ attend each other if and only if $\langle v_i, c_j \rangle / \langle c_j, v_i \rangle \in E'$. More formally, we use the following graph-guided masked attention matrix as the mask matrix $M$ in the equation 4:

$$M_{ij} = \begin{cases} 0 & \text{if } q_i \in \{[CLS], [SEP]\} \text{ or } q_i, k_j \in W \cup C \text{ or } \langle q_i, k_j \rangle \in E \cup E' \\ -\infty & \text{otherwise} \end{cases} \quad (6)$$

## 4.3 PRE-TRAINING TASKS

We describe three pre-training tasks used for pre-training GraphCodeBERT in this section. The first task is masked language modeling (Devlin et al., 2018) for learning representation from the source code. The second task is data flow edge prediction for learning representation from data flow, where we first mask some variables' data flow edges and then let GraphCodeBERT predict those edges. The last task is variable-alignment across source code and data flow for aligning representation between source code and data flow, which predicts where a variable is identified from.

**Masked Language Modeling** We follow Devlin et al. (2018) to apply masked language modeling (MLM) pre-training task. Specially, we sample randomly 15% of the tokens from the source code and paired comment. We replace them with a [MASK] token 80% of the time, with a random token 10% of the time, and leave them unchanged 10% of the time. The MLM objective is to predict original tokens of these sampled tokens, which has proven effective in previous works (Devlin et al., 2018; Liu et al., 2019; Feng et al., 2020). In particular, the model can leverage the comment context if the source code context is not sufficient to infer the masked code token, encouraging the model to align the natural language and programming language representations.

**Edge Prediction** To learn representation from data flow, we introduce a pre-training task of data flow edges prediction. The motivation is to encourage the model to learn structure-aware representation that encodes the relation of "where-the-value-comes-from" for better code understanding. Specially, we randomly sample 20% of nodes $V_s$ in data flow, mask direct edges connecting these sampled nodes by add an infinitely negative value in the mask matrix, and then predict these masked edges $E_{mask}$. Taking the variable $x^{11}$ in Figure 2 for an example, we first mask edges $\langle x^7, x^{11} \rangle$ and $\langle x^9, x^{11} \rangle$ in the graph and then let the model to predict these edges. Formally, the pre-training objective of the task is calculated as Equation 7, where $E_c = V_s \times V \cup V \times V_s$ is a set of candidates for edge prediction, $\delta(e_{ij} \in E)$ is 1 if $\langle v_i, v_j \rangle \in E$ otherwise 0, and the probability $p_{e_{ij}}$ of existing an edge from $i$-th to $j$-th node is calculated by dot product following a sigmoid function using representations of two nodes from GraphCodeBERT. To balance positive-negative ratio of examples, we sample negative and positive samples with the same number for $E_c$.

$$loss_{EdgePred} = -\sum_{e_{ij} \in E_c} [\delta(e_{ij} \in E_{mask}) log p_{e_{ij}} + (1 - \delta(e_{ij} \in E_{mask})) log(1 - p_{e_{ij}})] \quad (7)$$

**Node Alignment** To align representation between source code and data flow, we introduce a pre-training task of node alignment across source code and data flow, which is similar to data flow edge prediction. Instead of predicting edges between nodes, we predict edges between code tokens and nodes. The motivation is to encourage the model to align variables and source code according to data flow. Taking Figure 3 for an example, we first mask edges between the variable $x^{11}$ in data flow and code tokens, and then predict which code token the variable $x^{11}$ in data flow is identified from. As we can see, the model could predict that the variable $x^{11}$ is identified form the variable $x$ in the expression "return x" according to data flow information (i.e. the value of $x^{11}$ comes from $x^7$ or $x^9$).

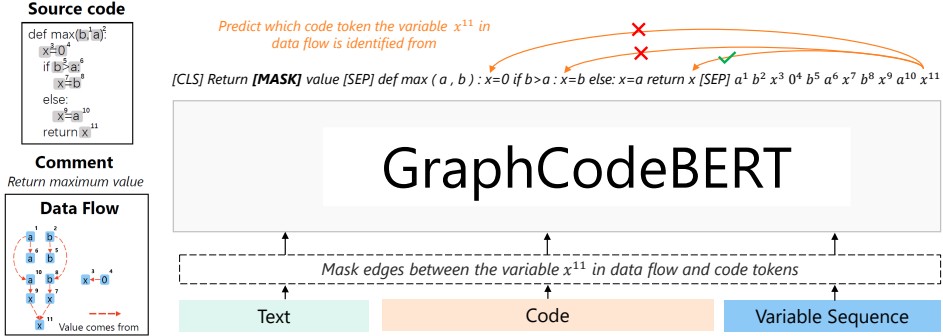

Figure 3: An example of the Node Alignment task.

Specially, we randomly sample 20% nodes $V_s^{'}$ in the graph, mask edges between code tokens and sampled nodes, and then predict masked edges $E_{mask}^{'}$. The pre-training objective of this task is similar to Equation 7, where $E_c^{'} = V_s^{'} \times C$ is a set of candidates for node alignment. Similarly, we also sample negative and positive samples with the same number for $E_c^{'}$.

$$loss_{NodeAlign} = - \sum_{e_{ij} \in E_c^{'}} [\delta(e_{ij} \in E_{mask}^{'})logp_{e_{ij}} + (1 - \delta(e_{ij} \in E_{mask}^{'}))log(1 - p_{e_{ij}})] \quad (8)$$

## 5 EXPERIMENTS

We evaluate our model on four downstream tasks, including code search, clone detection, code translation and code refinement. Detailed experimental settings can be found in the Appendix.

### 5.1 NATURAL LANGUAGE CODE SEARCH

Given a natural language as the input, the task aims to find the most semantically related code from a collection of candidate codes. We conduct experiments on the CodeSearchNet code corpus (Husain et al., 2019), which includes six programming languages. Different from the dataset and the setting used in the Husain et al. (2019), we filter low-quality queries by handcrafted rules and expand 1000 candidates to the whole code corpus, which is closer to the real-life scenario. We use Mean Reciprocal Rank (MRR) as our evaluation metric and report results of existing methods in the Table 1. We provide more details about the filtered dataset and also give results using the same setting of Husain et al. (2019) in the Appendix B.

| model | Ruby | Javascript | Go | Python | Java | Php | Overall |
|---|---|---|---|---|---|---|---|
| NBow | 0.162 | 0.157 | 0.330 | 0.161 | 0.171 | 0.152 | 0.189 |
| CNN | 0.276 | 0.224 | 0.680 | 0.242 | 0.263 | 0.260 | 0.324 |
| BiRNN | 0.213 | 0.193 | 0.688 | 0.290 | 0.304 | 0.338 | 0.338 |
| selfAtt | 0.275 | 0.287 | 0.723 | 0.398 | 0.404 | 0.426 | 0.419 |
| RoBERTa | 0.587 | 0.517 | 0.850 | 0.587 | 0.599 | 0.560 | 0.617 |
| RoBERTa (code) | 0.628 | 0.562 | 0.859 | 0.610 | 0.620 | 0.579 | 0.643 |
| CodeBERT | 0.679 | 0.620 | 0.882 | 0.672 | 0.676 | 0.628 | 0.693 |
| GraphCodeBERT | **0.703** | **0.644** | **0.897** | **0.692** | **0.691** | **0.649** | **0.713** |

Table 1: Results on code search. GraphCodeBERT outperforms other models significantly ($p < 0.01$).

All models calculate inner product of code and query encodings as relevance scores to rank candidate codes. We follow Husain et al. (2019) to implement four methods as baselines in the first group to obtain the encodings, including bag-of-words, convolutional neural network, bidirectional recurrent neural network, and multi-head attention. The second group is the results of pre-trained models. Roberta (Liu et al., 2019) is a pre-trained model on text corpus with MLM learning objective, while **RoBERTa (code)** is pre-trained only on code. **CodeBERT** (Feng et al., 2020) is pre-trained

on code-text pairs with MLM and replaced token detection learning objectives. As we can see, **GraphCodeBERT** that leverages code structure for pre-training brings a 2% gain of MRR, achieving the state-of-art performance. We also conducted t-test between our GraphCodeBERT and other baselines, and the results show the improvements are significant with $p < 0.01$.

## 5.2 CODE CLONE DETECTION

Code clones are multiple code fragments that output similar results when given the same input. The task aims to measure the similarity between two code fragments, which can help reduce the cost of software maintenance and prevent bugs. We conduct experiments on the BigCloneBench dataset (Svajlenko et al., 2014) and report results in the Table 2.

**Deckard** (Jiang et al., 2007) is to compute vectors for structural information within ASTs and then a Locality Sensitive Hashing (LSH) (Datar et al., 2004) is used to cluster similar vectors for detection. **RtvNN** (White et al., 2016) trains a recursive autoencoder to learn representations for AST. **CDLH** (Wei & Li, 2017) learn representations of code fragments via AST-based LSTM and hamming distance is used to optimize the distance between the vector representation of AST pairs.

**ASTNN** Zhang et al. (2019) uses RNNs to encode AST subtrees for statements, then feed the encodings of all statement trees into an RNN to learn representation for a program. **FA-AST-GMN** (Wang et al., 2020) uses GNNs over a flow-augmented AST to leverages explicit control and data flow information for code clone detection. Results show that our **GraphCodeBERT** that leverages code structure information significantly outperforms other methods with $p < 0.01$, which demonstrates the effectiveness of our pre-trained model for the task of code clone detection.

| Model | Precision | Recall | F1 |
|---|---|---|---|
| Deckard | 0.93 | 0.02 | 0.03 |
| RtvNN | 0.95 | 0.01 | 0.01 |
| CDLH | 0.92 | 0.74 | 0.82 |
| ASTNN | 0.92 | 0.94 | 0.93 |
| FA-AST-GMN | 0.96 | 0.94 | 0.95 |
| RoBERTa (code) | 0.960 | 0.955 | 0.957 |
| CodeBERT | 0.964 | 0.966 | 0.965 |
| GraphCodeBERT | **0.973** | **0.968** | **0.971** |

Table 2: Results on code clone detection. GraphCodeBERT outperforms other pre-trained methods significantly ($p < 0.01$).

## 5.3 CODE TRANSLATION

Code translation aims to migrate legacy software from one programming language in a platform to another. Following Nguyen et al. (2015) and Chen et al. (2018), we conduct experiments on a dataset crawled from the same several open-source projects as them and report results in the Table 3.

The **Naive** method is directly copying the source code as the translation result. **PBSMT** is short for phrase-based statistical machine translation (Koehn et al., 2003), and has been exploited in previous works (Nguyen et al., 2013; Karaivanov et al., 2014). As for the **Transformer**, we use the same number of layers and hidden size as pre-trained models. To leverage the pre-trained models for translation, we initialize the encoder with pre-trained models and randomly initialize parameters of the decoder and the source-to-target attention. Results show that the models initialized with pre-trained models (i.e the second group) significantly outperform PBSMT and Transformer models. Among them, **GraphCodeBERT** achieves state-of-art performance, which demonstrates the effectiveness of our model for code translation.

| Method | Java→C# | | C#→Java | |
|---|---|---|---|---|
| | BLEU | Acc | BLEU | Acc |
| Naive | 18.54 | 0.0 | 18.69 | 0.0 |
| PBSMT | 43.53 | 12.5 | 40.06 | 16.1 |
| Transformer | 55.84 | 33.0 | 50.47 | 37.9 |
| RoBERTa (code) | 77.46 | 56.1 | 71.99 | 57.9 |
| CodeBERT | 79.92 | 59.0 | 72.14 | 58.0 |
| GraphCodeBERT | **80.58** | **59.4** | **72.64** | **58.8** |

Table 3: Results on code translation. GraphCodeBERT outperforms other models significantly ($p < 0.05$).

## 5.4 CODE REFINEMENT

Code refinement aims to automatically fix bugs in the code, which can contribute to reducing the cost of bug-fixes. We use the dataset released by Tufano et al. (2019) and report results in the Table 4.

The **Naive** method directly copies the buggy code as the refinement result. For the **Transformer**, we use the same number of layers and hidden size as the pre-trained models. Same as the Section 5.3, we initialize the encoder with pre-trained models and randomly initialize parameters of the decoder and the source-to-target attention. Then we use the training data to fine-tune the whole model. In the table, we see that the **Transformer** significantly outperforms **LSTM**. Results in the second group shows that pre-trained models outperform Transformer models further, and **GraphCodeBERT** achieves better performance than other pre-trained models on both datasets, which shows leveraging code structure information are helpful to the task of code refinement.

| Method | small | | medium | |
|---|---|---|---|---|
| | BLEU | Acc | BLEU | Acc |
| Naive | 78.06 | 0.0 | 90.91 | 0.0 |
| LSTM | 76.76 | 10.0 | 72.08 | 2.5 |
| Transformer | 77.21 | 14.7 | 89.25 | 3.7 |
| RoBERTa (code) | 77.30 | 15.9 | 90.07 | 4.1 |
| CodeBERT | 77.42 | 16.4 | 91.07 | 5.2 |
| GraphCodeBERT | **80.02** | **17.3** | **91.31** | **9.1** |

Table 4: Results on code refinement.

## 5.5 MODEL ANALYSIS

**Ablation Study**    We conduct ablation study on the task of natural language code search to understand various components in our approach impact overall performance. We remove two pre-training tasks and data flow, respectively, to analyze their contribution. Table 5 shows that the overall performance drops from 71.3% to 70.3%~70.7% when removing Node Alignment and Edge Prediction pre-training tasks, respectively, which reveals the importance of two structure-aware pre-training tasks. After ablating the data flow totally, we can see that the performance drops from 71.3% to 69.3%, which means leveraging data flow to learn code representation could improve GraphCodeBERT.

| Methods | Ruby | Javascript | Go | Python | Java | Php | Overall |
|---|---|---|---|---|---|---|---|
| GraphCodeBERT | **0.703** | **0.644** | **0.897** | **0.692** | **0.691** | **0.649** | **0.713** |
| -w/o EdgePred | 0.701 | 0.632 | 0.894 | 0.687 | 0.688 | 0.640 | 0.707 |
| -w/o NodeAlign | 0.685 | 0.635 | 0.887 | 0.682 | 0.690 | 0.640 | 0.703 |
| -w/o Data Flow | 0.679 | 0.620 | 0.882 | 0.672 | 0.676 | 0.628 | 0.693 |

Table 5: Ablation study on natural language code search

**Node-vs. Token-level Attention**    Table 6 shows how frequently a special token $[CLS]$ that is used to calculate probability of correct candidate attends to code tokens (Codes) and variables (Nodes). We see that although the number of nodes account for 5%~20%, attentions over nodes overwhelm node/code ratio (around 10% to 32%) across all programming languages. The results indicate that data flow plays an important role in code understanding process and the model pays more attention to nodes in data flow than code tokens.

| | Ruby | Javascript | Go | Python | Java | Php |
|---|---|---|---|---|---|---|
| Codes/Nodes | 90.1/9.9 | 94.6/5.4 | 95.0/5.03 | 80.6/19.4 | 93.2/6.8 | 87.5/12.5 |
| $[CLS] \rightarrow$Codes/Nodes | 82.3/17.7 | 89.7/10.3 | 91.0/9.0 | 67.7/32.3 | 87.8/12.2 | 79.4/20.6 |

Table 6: Attention distribution (%) between code tokens (codes) and variables (nodes) across different programming language on natural language code search test sets. The first row is the ratio of the number of code tokens to nodes, and the second row is attention distribution of $[CLS]$ token.

**Comparison between AST and Data Flow**    Figure 4 shows MRR score with respect to input sequence length on the validation dataset of Ruby programming language for the task of code search. **AST Pre-order Traversal** regards AST as a sequence by linearizing all AST nodes using pre-order traversal algorithm. **AST Subtree Masking** regards AST as a tree and introduce subtree masking (Nguyen et al., 2019) for self-attention of the Transformer. In subtree masking, each node-query in AST attends only to its own subtree descendants, and each leaf-query only attends to leaves of AST. Transformer has a self-attention component with $O(n^2)$ time and memory complexity where $n$ is the input sequence length, and thus is not efficient to scale to long inputs.

We observe that injecting AST even hurts the performance when the sequence length is short (e.g. shorter than 128), while Graph-CodeBERT consistently brings performance boost on varying sequence length and obtains better MRR score than AST-based methods. The main reason is that data flow is less complex and the number of nodes account for $5\% \sim 20\%$ (see Table 6), which does not bring an unnecessarily deep hierarchy of AST and makes the model more accurate and efficient.

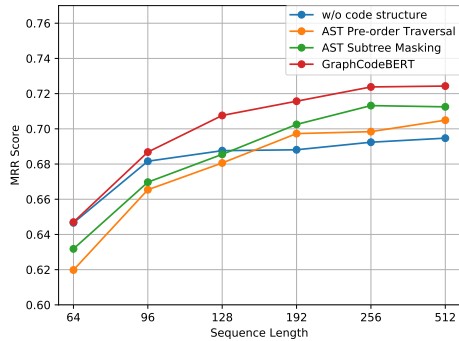

Figure 4: MRR score on the validation dataset of Ruby for code search with varying length of input sequence.

**Case Study**  We also give a case study to demonstrate that data flow would enhance the code understanding process. Given a source code and a comment, we use GraphCodeBERT with and without data flow to predict whether the comment correctly describes the source code. Results are given in Figure 5. We can see that both models make correct prediction in the original example, where the threshold is 0.5 (left panel). To study the code understanding ability of models, we change the source code (center panel) and the comment (right panel), respectively. Although we make a small change on the source code ($return\ a \rightarrow return\ b$) and the comment ($sum\ value \rightarrow mean\ value$), the semantic of the source code and the comment are completely different and corresponding gold labels change from 1 to 0. As we can see in the figure, GraphCodeBERT without using data flow fails these tests and still outputs high probability for negative examples. After leveraging data flow, GraphCodeBERT better understands the semantic of source code and makes correct predictions on all tests, which demonstrates that data flow could improve the code understanding ability of the model.

| | Unchanged | Code: $return\ a \rightarrow return\ b$ | NL: $sum\ value \rightarrow mean\ value$ |
|---|---|---|---|
| Input | **NL:**
Return sum value of an array
**Code:**
import numpy as np
def f(array):
    a=np.sum(array)
    b=np.mean(array)
    return a | **NL:**
Return sum value of an array
**Code:**
import numpy as np
def f(array):
    a=np.sum(array)
    b=np.mean(array)
    return **b** | **NL:**
Return **mean** value of an array
**Code:**
import numpy as np
def f(array):
    a=np.sum(array)
    b=np.mean(array)
    return a |
| Label | 1 | 0 | 0 |
| Prediction | **GraphCodeBERT:** 0.6563 (1)
**GraphCodeBERT:** 0.8728 (1)
**(w/o Data Flow)** | **GraphCodeBERT:** 0.4615 (0)
**GraphCodeBERT:** 0.8608 (1)
**(w/o Data Flow)** | **GraphCodeBERT:** 0.2884 (0)
**GraphCodeBERT:** 0.9048 (1)
**(w/o Data Flow)** |

Figure 5: We take a comment and a source code as the input (first row), and use GraphCodeBERT with and without data flow to predict the probability of the source code matching the comment (third row). The label is 1 if the comment correctly describes the source code otherwise 0 (second row).

## 6    CONCLUSION

In this paper, we present GraphCodeBERT that leverages data flow to learn code representation. To the best of our knowledge, this is the first pre-trained model that considers code structure for pre-training code representations. We introduce two structure-aware pre-training tasks and show that GraphCodeBERT achieves state-of-the-art performance on four code-related downstream tasks, including code search, clone detection, code translation and code refinement. Further analysis shows that code structure and newly introduced pre-training tasks boost the performance. Additionally, case study in the task of code search shows that applying data flow in the pre-trained model improves code understanding.

ACKNOWLEDGMENTS

Daya Guo and Jian Yin are supported by the Research Foundation of Science and Technology Plan Project in Guangdong Province (2017B030308007).

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

## A  PRE-TRAINING DETAILS

GraphCodeBERT includes 12 layers Transformer with 768 dimensional hidden states and 12 attention heads. For fair comparison, we use the same dataset as CodeBERT (Feng et al., 2020) to pretrain our model. The dataset is the CodeSearchNet dataset[3] (Husain et al., 2019), which includes 2.3M functions with document pairs for six programming languages. We train the model on two DGX-2 machines, each having 16 NVIDIA Tesla V100 with 32GB memory. We set the max length of sequences and nodes as 512 and 128, respectively. We use the Adam optimizer to update model parameters with 1,024 batch size and 2e-4 learning rate. To accelerate the training process, we adopt the parameters of CodeBERT released by Feng et al. (2020) to initialize the model. The model is trained with 200K batches and costs about 83 hours.

At each iteration, we alternate EdgePred and NodeAlign objectives in combination with MLM to pre-train the model. And we follow Lample & Conneau (2019) to sample each batch from the same programming language according to a multinomial distribution with probabilities $\{q_i\}_{i=1...N}$, where $n_i$ is number of examples for $i$-th programming language and $\alpha$=0.7. Sampling with this distribution could alleviates the bias towards high-resource languages.

$$q_i = \frac{p_i^{\alpha}}{\sum_N^{j=1} p_j^{\alpha}} \; with \; p_i = \frac{n_i}{\sum_N^{k=1} n_k} \tag{9}$$

## B  NATURAL LANGUAGE CODE SEARCH

Given a natural language as the input, code search aims to find the most semantically related code from a collection of candidate codes. We conduct experiments on the CodeSearchNet code corpus (Husain et al., 2019) and follow Husain et al. (2019) to take the first paragraph of the documentation as the query for the corresponding function. However, we observe that some queries contain content unrelated to the code, such as a link "http://..." that refers to external resources. Therefore, we filter following examples to improve the quality of the dataset.

(1) Examples whose code could not be parsed into abstract syntax tree.

(2) Examples whose query tokens number is shorter than 3 or larger than 256.

(3) Examples whose query contains special tokens such as "http://".

(4) Examples whose query is empty or not written in English.

---

[3]https://github.com/github/CodeSearchNet

Different from the setting of Husain et al. (2019), the answer of each query is retrieved from the whole development and testing code corpus instead of 1,000 candidate codes. We list data statistics about the filtered dataset in Table 7.

| Code Search | Training examples | Dev queries | Testing queries | Candidate codes |
|---|---|---|---|---|
| Go | 167,288 | 7,325 | 8,122 | 28,120 |
| Java | 164,923 | 5,183 | 10,955 | 40,347 |
| JavaScript | 58,025 | 3,885 | 3,291 | 13,981 |
| PHP | 241,241 | 12,982 | 14,014 | 52,660 |
| Python | 251,820 | 13,914 | 14,918 | 43,827 |
| Ruby | 24,927 | 1,400 | 1,261 | 4,360 |

Table 7: Data statistics about the filtered dataset. For each query in the development and testing sets, the answer is retrieved from the whole candidate codes (i.e. the last row).

We use GraphCodeBERT to separately encode query and source code with data flow, and calculate inner product of their representations of the special token $[CLS]$ as relevance scores to rank candidate codes. In the fine-turning step, we set the learning rate as 2e-5, the batch size as 32, the max sequence length of queries and codes as 128 and 256, and the max number of nodes as 64. We use the Adam optimizer to update model parameters and perform early stopping on the development set.

We also report the results using the same setting of Husain et al. (2019) in Table 8. In this setting, models are required to retrieve an answer for a query from 1000 candidates. The results show that GraphCodeBERT also achieves the state-of-the-art performance.

| model | Ruby | Javascript | Go | Python | Java | Php | Overall |
|---|---|---|---|---|---|---|---|
| NBow | 0.429 | 0.461 | 0.641 | 0.581 | 0.514 | 0.484 | 0.518 |
| CNN | 0.245 | 0.352 | 0.627 | 0.571 | 0.527 | 0.529 | 0.475 |
| BiRNN | 0.084 | 0.153 | 0.452 | 0.321 | 0.287 | 0.251 | 0.258 |
| selfAtt | 0.365 | 0.451 | 0.681 | 0.692 | 0.587 | 0.601 | 0.563 |
| RoBERTa | 0.625 | 0.606 | 0.820 | 0.809 | 0.666 | 0.658 | 0.697 |
| RoBERTa (code) | 0.661 | 0.640 | 0.819 | 0.844 | 0.721 | 0.671 | 0.726 |
| CodeBERT | 0.693 | 0.706 | 0.840 | 0.869 | 0.748 | 0.706 | 0.760 |
| GraphCodeBERT | **0.732** | **0.711** | **0.841** | **0.879** | **0.757** | **0.725** | **0.774** |

Table 8: Results on natural language code search using the setting of Husain et al. (2019).

## C  CODE CLONE DETECTION

Code clone detection aims to measure the similarity between two code fragments. We use Big-CloneBench dataset (Svajlenko et al., 2014), which contains over 6,000,000 true clone pairs and 260,000 false clone pairs from 10 different functionalities. We follow the settings in Wei & Li (2017), discarding code fragments without any tagged true and false clone pairs and using 9,134 remaining code fragments. Finally, the dataset provided by Wang et al. (2020) includes 901,724/416,328/416,328 examples for training/validation/testing. We treat the task as a binary classification to fine-tune Graph-CodeBERT, where we use source code and data flow as the input. The probability of true clone is calculated by dot product from the representation of $[CLS]$. In the fine-turning step, we set the learning rate as 2e-5, the batch size as 16, the max sequence length as 512 the max number of nodes as 128. We use the Adam optimizer to update model parameters and tune hyper-parameters and perform early stopping on the development set.

We give a case of the GraphCodeBERT output for this task in Figure 6. In this example, two Java source codes both download content from a given URL and convert the type of the content into string type. Therefore, two codes are semantically similar since they output similar results when given the same input. As we can see, our model gives a high score for this case and the pair is classified as true clone pair.

**Input:** Two source codes          **Output:** Semantically similar (score: 0.983)

```
protected String downloadURLtoString(URL url) throws IOException
{
    BufferedReader in = new BufferedReader(new
                        InputStreamReader(url.openStream()));
    StringBuffer sb = new StringBuffer(100 * 1024);
    String str;
    while ((str = in.readLine()) != null) {
        sb.append(str);
    }
    in.close();
    return sb.toString();
}
```

```
public static String fetchUrl(String urlString)
{
    try {
        URL url = new URL(urlString);
        BufferedReader reader = new BufferedReader(new
                            InputStreamReader(url.openStream()));
        String line = null;
        StringBuilder builder = new StringBuilder();
        while ((line = reader.readLine()) != null) {
            builder.append(line);
        }
        reader.close();
        return builder.toString();
    } catch (MalformedURLException e) {
    } catch (IOException e) {
    }
    return "";
}
```

Figure 6: A case of GraphCodeBERT output for the code clone detection task.

## D  CODE TRANSLATION

Code translation aims to migrate legacy software from one programming language in a platform to another. We conduct experiments on a dataset crawled from the same several open-source projects as Nguyen et al. (2015) and Chen et al. (2018), i.e. Lucene[4], POI[5], JGit[6] and Antlr[7]. We do not use Itext[8] and JTS[9] as they do because of the license problem. Those projects have both Java and C# implementation. We pair the methods in the two languages based on their file names and method names. After removing duplication and methods with null function body, the total number of method pairs is 11,800, and we split 500 pairs from them as the development set and another 1,000 pairs for test. To demonstrate the effectiveness of GraphCodeBERT on the task of code translation, we adopt various pre-trained models as encoders and stay hyperparameters consistent. We set the learning rate as 1e-4, the batch size as 32, the max sequence length as 256 and the max number of nodes as 64. We use the Adam optimizer to update model parameters and tune hyper-parameters and perform early stopping on the development set.

We give a case of the GraphCodeBERT output for this task in Figure 7. In this example, the model successfully translates a piece of Java code into its C# version. The differences include the type name (from "boolean" to "bool") and the usage of getting a string value of a bool variable (from "String.valueOf(b)" to "b.ToString()").

**Input:** A Java method          **Output:** Its C# version

```
public void print(boolean b)
{
    print(String.valueOf(b));
}
```
→
```
public void print(bool b)
{
    print(b.ToString());
}
```

Figure 7: A case of GraphCodeBERT output for the code translation task.

---

[4]http://lucene.apache.org/

[5]http://poi.apache.org/

[6]https://github.com/eclipse/jgit/

[7]https://github.com/antlr/

[8]http://sourceforge.net/projects/itext/

[9]http://sourceforge.net/projects/jts-topo-suite/

# E    CODE REFINEMENT

Code refinement aims to automatically fix bugs in the code. We use the dataset released by Tufano et al. (2019). The source is buggy Java functions while the target is the according fixed ones. Almost all the names of variables and custom methods are normalized. The dataset contains two subsets based on the code length. For the *small* dataset, the numbers of training, development and test samples are 46,680, 5,835 and 5,835. For the *medium* dataset, the numbers are 52,364, 6,545 and 6,545. We also use the sequence-to-sequence Transformer model to conduct the experiments. In the fine-tuning step, we adopt various pre-trained models as encoders. We set the learning rate as 1e-4, the batch size as 32, the max sequence length as 256 and the max number of nodes as 64. We use the Adam optimizer to update model parameters and perform early stopping on the development set.

We give two cases of the GraphCodeBERT output for this task in Figure 8. In the first example, the model successfully fixes the operation bug (from "*" to "+") to match the function name "add". In the second case, the source function and type names are normalized. The return type of this function is "void" but the buggy code gives a return value. Our model successfully removes the "return" word so that the return type of the function matches its declaration.

**Input:** A buggy Java method
```
public int add ( int a , int b )
{
      return a * b ;
}
```
```
public void METHOD_1 ( TYPE_1 c )
{
      return VAR_1 . remove ( c ) ;
}
```

**Output:** The fixed one
```
public int add ( int a , int b )
{
      return a + b ;
}
```
```
public void METHOD_1 ( TYPE_1 c )
{
      VAR_1 . remove ( c ) ;
}
```

Figure 8: Two cases of GraphCodeBERT output for the code refinement task.

# F    CASE STUDY

## F.1    NATURAL LANGUAGE CODE SEARCH

We give a case study to illustrate retrieved results by GraphCodeBERT on the natural language code search task, with a comparison to CodeBERT and RoBERTa (code) models. Two examples are given in Figure 9 and we can see that GraphCodeBERT successfully retrieves correct source codes for given queries on both examples. As we can see in the first case, incorporating data flow will help Graph-CodeBERT better understand the complicated expression "[(k, v) for k, v in self.items() if v is not self.EMPTY]" by leveraging dependency relation among variables in data flow graph. In the second case, the terminology "%Y-%m-%d" in Python program language is a format of date time. GraphCodeBERT and CodeBERT both successfully search the correct function. Compared with RoBERTa (code), the second case shows that utilizing natural language descriptions for pre-training helps models do better semantic matching between source codes and queries on the code search task.

## F.2    CODE CLONE DETECTION

We give a case study to compare GraphCodeBERT with CodeBERT and RoBERTa (code) models on code clone detection task. An example is shown in Figure 10. The first source code is to return the HTML content from a given URL, while the second source code is to return the last line from a fixed URL "http://kmttg.googlecode.com/svn/trunk/version". Their semantics are not similar due to their different outputs. Data flow could help GraphCodeBERT better understand that the return value "pageHTML" in first source code comes from "pageHTML.append(line); pageHTML.append("\r\n");" instead of "bufferedWriter.write(pageHTML.toString());" and the return value "version" in the second source code comes from "version = inputLine" or "version = null;". Although two source codes are highly overlapped (marked in yellow), GraphCodeBERT successfully predict the gold label compared with other models without data flow.

**Case 1**
**Query:** Return copy of instance, omitting entries that are EMPTY
**Gold Source Code:**
```
def defined_items(self):
    return self.__class__(
        [(k, v) for k, v in self.items() if v is not self.EMPTY], is_empty=False
    )
```

**Search Results (Top1)**
**GraphCodeBERT:**
```
def defined_items(self):
    return self.__class__(
        [(k, v) for k, v in self.items() if v is not self.EMPTY], is_empty=False
    )
```

**CodeBERT:**
```
def copy(self):
    context = CLIContext()
    for item in dir(self):
        if item[0] != '_' and item not in ('copy', 'write_headers'):
            setattr(context, item, getattr(self, item))
    return context
```

**RoBERTa (code):**
```
def copy(self):
    x = self.to_dict()
    x.pop(self._pkey)
    return self.__class__(**x)
```

**Case 2**
**Query:** Fast %Y-%m-%d parsing
**Gold Source Code:**
```
def parse_date(s):
    try:
        return datetime.date(int(s[:4]), int(s[5:7]), int(s[8:10]))
    except ValueError:
        return datetime.datetime.strptime(s, '%d %B %Y').date()
```

**Search Results (Top1)**
**GraphCodeBERT:**
```
def parse_date(s):
    try:
        return datetime.date(int(s[:4]), int(s[5:7]), int(s[8:10]))
    except ValueError:
        return datetime.datetime.strptime(s, '%d %B %Y').date()
```

**CodeBERT:**
```
def parse_date(s):
    try:
        return datetime.date(int(s[:4]), int(s[5:7]), int(s[8:10]))
    except ValueError:
        return datetime.datetime.strptime(s, '%d %B %Y').date()
```

**RoBERTa (code):**
```
def parse(self, hcl, canonicalize=False):
    return self.request("parse", json={"JobHCL": hcl, "Canonicalize":
canonicalize}, method="post", allow_redirects=True).json()
```

Figure 9: Two examples on code search task and retrieved results from different models.

**Source code 1:**
```
private String getHTML(String pageURL, String encoding, String dirPath)
throws IOException {
    StringBuilder pageHTML = new StringBuilder();
    HttpURLConnection connection = null;
    try {
        URL url = new URL(pageURL);
        connection = (HttpURLConnection) url.openConnection();
        connection.setRequestProperty("User-Agent", "MSIE 7.0");
        connection.connect();
        BufferedReader br = new BufferedReader(new
InputStreamReader(connection.getInputStream(), encoding));
        String line = null;
        while ((line = br.readLine()) != null) {
            pageHTML.append(line);
            pageHTML.append("\r\n");
        }
    } catch (Exception e) {
        e.printStackTrace();
    } finally {
        connection.disconnect();
    }
    if (dirPath != null) {
        File file = new File(dirPath);
        BufferedWriter bufferedWriter = new BufferedWriter(new
FileWriter(file));
        bufferedWriter.write(pageHTML.toString());
        bufferedWriter.close();
    }
    return pageHTML.toString();
}
```

**Source code 2:**
```
private static String getVersion() {
    debug.print("");
    String version = null;
    String version_url =
"http://kmttg.googlecode.com/svn/trunk/version";
    try {
        URL url = new URL(version_url);
        URLConnection con = url.openConnection();
        BufferedReader in = new BufferedReader(new
InputStreamReader(con.getInputStream()));
        String inputLine;
        while ((inputLine = in.readLine()) != null) version = inputLine;
        in.close();
    } catch (Exception ex) {
        version = null;
    }
    return version;
}
```

**Gold Label:**
No semantically similar

**Prediction:**
**GraphCodeBERT:**
No semantically similar

**CodeBERT:**
semantically similar

**RoBERTa (code):**
semantically similar

Figure 10: An examples on code clone detection task and model prediction from different models. Overlapped code snippets between two source codes are marked in yellow.

## F.3 CODE TRANSLATION AND CODE REFINEMENT

We give a case study to compare GraphCodeBERT with Transformer without using data flow on code generation tasks, including code translation and code refinement. We list three cases in Table 9 and Table 10, respectively. [src] represents the source input, [ref] represents the reference, [sys] represents Transformer without data flow and [ours] represents GraphCodeBERT. We can see that the Transformer ([sys]) baseline makes several mistakes, including repeating tokens, logic errors and syntax errors, while GraphCodeBERT ([ours]) as a encoder could improve the generation.

| | Case1: Transformer outputs repeating tokens |
|---|---|
| [src] | public static final WeightedTerm[] getTerms(Query query){return getTerms(query,false);} |
| [ref] | public static WeightedTerm[] GetTerms(Query query){return GetTerms(query, false);} |
| [sys] | public static WeightedTerm[] GetTerms(Query query){return GetTerms(false, new static static static static static static WeightTerms);} |
| [ours] | public static WeightedTerm[] GetTerms(Query query){return GetTerms(query, false);} |
| | Case2: Transformer outputs codes with severe logic and syntax errors |
| [src] | public long skip(long n){int s = (int) Math.min(available(), Math.max(0, n));ptr += s;return s;} |
| [ref] | public override long Skip(long n){int s = (int)Math.Min(Available(), Math.Max(0, n)); ptr += s;return s;} |
| [sys] | public override long Skip(long n){int s = Math.Min(n) == 0 ? Math.Min(00.0 : Math.Min(n, s.Length);return s;} |
| [ours] | public override long Skip(long n){int s = (int)Math.Min(Available(), Math.Max(0, n)); ptr += s;return s;} |
| | Case3: Transformer uses the wrong variable as a parameter. |
| [src] | public UnbufferedCharStream(int bufferSize){n = 0;data = new int[bufferSize];} |
| [ref] | public UnbufferedCharStream(int bufferSize){n = 0;data = new int[bufferSize];} |
| [sys] | public UnbufferedCharStream(int bufferSize){data = new int[data];} |
| [ours] | public UnbufferedCharStream(int bufferSize){n = 0;data = new int[bufferSize];} |

Table 9: Three examples that translate from Java to C# programming language on code translation task. [src] represents the source input, [ref] represents the reference, [sys] represents Transformer without data flow and [ours] represents GraphCodeBERT.

| | Case1: Transformer adds redundant parameters (android.view.View view) |
|---|---|
| [src] | public void METHOD_1 ( ) { android.content.Intent VAR_1 = new android.content.Intent ( VAR_2 ) ; METHOD_2 ( VAR_1 , 0 ) ; android.content.Intent i = new android.content.Intent ( this , VAR_3 class ) ; METHOD_3 ( i ) ; } |
| [ref] | public void METHOD_1 ( ) { android.content.Intent VAR_1 = new android.content.Intent ( VAR_2 ) ; METHOD_2 ( VAR_1 , 0 ) ; } |
| [sys] | public void METHOD_1 ( android.view.View view ) { android.content.Intent VAR_1 = new android.content.Intent ( VAR_2 ) ; METHOD_2 ( VAR_1 , 0 ) ; } |
| [ours] | public void METHOD_1 ( ) { android.content.Intent VAR_1 = new android.content.Intent ( VAR_2 ) ; METHOD_2 ( VAR_1 , 0 ) ; } |
| | Case2: Transformer outputs codes with severe logic or irrelevant codes |
| [src] | public java.util.Date METHOD_1 ( ) { return VAR_1 . METHOD_1 ( ) . METHOD_2 ( ) ; } |
| [ref] | public java.util.Date METHOD_1 ( ) { if ( ( VAR_1 . METHOD_1 ( ) ) != null ) { return VAR_1 . METHOD_1 ( ) . METHOD_2 ( ) ; } else { return null ; } } |
| [sys] | public java.util.Date METHOD_1 ( ) { if ( ( VAR_1 ) == null ) { return new java.util.Date ( ) ; } return VAR_1 . METHOD_1 ( ) . METHOD_2 ( ) ; } |
| [ours] | public java.util.Date METHOD_1 ( ) { if ( ( VAR_1 . METHOD_1 ( ) ) != null ) { return VAR_1 . METHOD_1 ( ) . METHOD_2 ( ) ; } else { return null ; } } |
| | Case3: Transformer makes no change |
| [src] | public java.lang.String METHOD_1 ( TYPE_1 VAR_1 ) { if ( VAR_1 == null ) return null ; return VAR_1 . METHOD_2 ( ) . getText ( ) ; } |
| [ref] | public java.lang.String METHOD_1 ( TYPE_1 VAR_1 ) { return VAR_1 . METHOD_2 ( ) . getText ( ) ; } |
| [sys] | public java.lang.String METHOD_1 ( TYPE_1 VAR_1 ) { if ( VAR_1 == null ) return null ; return VAR_1 . METHOD_2 ( ) . getText ( ) ; } |
| [ours] | public java.lang.String METHOD_1 ( TYPE_1 VAR_1 ) { return VAR_1 . METHOD_2 ( ) . getText ( ) ; } |

Table 10: Three examples on code refinement task. [src] represents the source input, [ref] represents the reference, [sys] represents Transformer without data flow and [ours] represents GraphCodeBERT.

## G   ERROR ANALYSIS

We also conduct error analysis and summary two main classes of errors for both code understanding and generation tasks.

Figure 11 gives three error cases of GraphCodeBERT on the natural language code search task. We observe that GraphCodeBERR mainly fails to retrieve those source code that involves functions of the library like "tf" (Tensorflow) in the first case and " GoogleCloudStorageHook" in the second case. It's difficult for GraphCodeBERR to understand meanings of APIs like "tf.io.read_file" and "tf.image.decode_image" without relevant information. A potential direction to mitigate the problem is to incorporate definitions of the library. The other major problem is that there are some terminologies like "unistr" in the query (corresponding to "decode('utf-8')" in Python code) in third case. Incorporating more text-code pairs for pre-training might alleviate this problem.

As for the code generation task, Table 11 shows two cases of GraphCodeBERT on the code translation task. We find that the major problems include semantic errors like identifiers from nowhere in the first case and syntax errors like missing a "}" symbol before "return n" in the second case. This problem might be mitigated by incorporating a dedicated decoder that takes into account grammar of programming languages and different generation paradigm like generating a sequence of production rules (Yin & Neubig, 2017; Guo et al., 2018; 2019) in a context-free grammar manner.

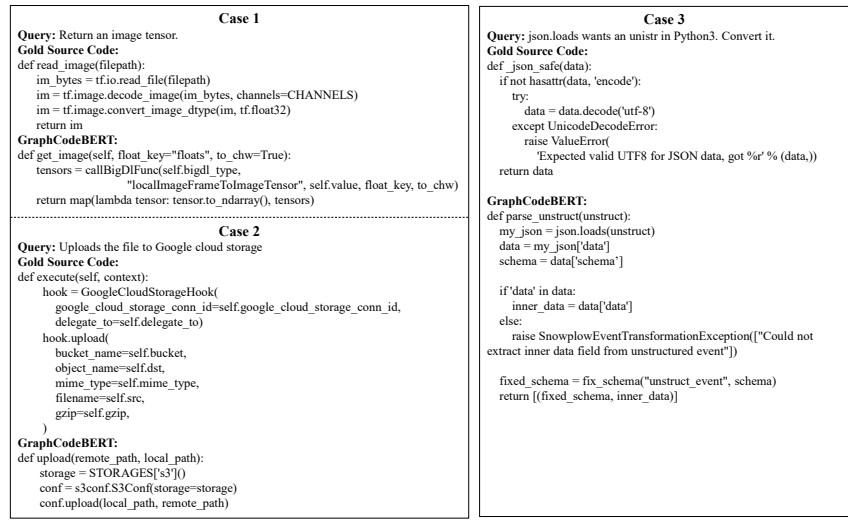

Figure 11: Error cases of GraphCodeBERT on the natural language code search.

| Case1: semantic error – identifiers from nowhere. | |
|---|---|
| [src] | public String toString() {return getKey() + ": " + getValue(); } |
| [ref] | public override string ToString(){return GetKey() + ": " + GetValue();} |
| [ours] | public override string ToString(){return Name + ": " + GetValue();} |
| Case2: syntax errors – missing a "}" before "return n") | |
| [src] | public static int numNonnull(Object[] data) {int n = 0;if ( data == null ) return n; for (Object o : data) {if ( o!=null ) n++;}return n;} |
| [ref] | public static int NumNonnull(object[] data){int n = 0;if (data == null){return n;} foreach (object o in data){if (o != null){n++;}}return n;} |
| [ours] | public static int NumNonNull(object[] data){int n = 0;if (data == null){return n;} foreach (object o in data){if (o != null){n++;}return n;} |

Table 11: Error cases of GraphCodeBERT on the code translation task. [src] represents the source input, [ref] represents the reference and [ours] represents GraphCodeBERT.

