# OpenReview forum: "GraphCodeBERT: Pre-training Code Representations with Data Flow"
_ICLR.cc/2021/Conference — ICLR 2021 Poster_

### Official Review · AnonReviewer4 · 2020-10-25
**AnonReviewer4**

**Rating:** 6
**Confidence:** 3

**Review:**

This paper proposes GraphCodeBERT as a Transformer-based pretrained model for programming language that incorporates data flow information in the graph representation of variables in the code. The data flow graph encodes the structure of variables based on “where-the-value-comes-from” from the AST parse. The pretrained model is jointly trained on the code, the natural language comment of the code, and the data flow graph of the code. In addition to the Masked Language Modeling objective, two new pretraining objectives are proposed including predicting the edge of the data flow graph and predicting the alignment of variables between data flow graph and code. The graph-guided masked attention is used such that the attention can only occur if two variables have an edge in the data flow graph or there is an alignment between data flow graph and code. The experiments show that GraphCodeBERT can deliver improvements on Natural Language Code Search, Code Clone Detection, Code Translation, and Code Refinement.

Strengths

- The approach is well motivated, and I totally agree that we should consider the code structure for a pretrained code language model.

- There are consistent, though small, improvements on all four tasks.

- The case study and ablation study are very helpful to understand why GraphCodeSearch is better than other models with better representation of code semantics on the Code Search task.

Weakness

- This paper uses the data flow graph and claims this is better than AST because AST has an "unnecessarily deep hierarchy". However, as the data flow graph is extracted from AST, data flow contains only partial information of AST, especially the information of “where-the-value-comes-from”, and throws out other useful information such as function name, condition control, operator. It is not clear to me if this design choice is a good one.

- The experiments should have a comprehensive comparison using AST and the data flow graph to answer the questions such as: What is the performance of using AST instead of the data flow graph? What information in AST is useful for which tasks? In addition to “where-the-value-comes-from” information, what other information shall we incorporate into a pretrained language model for code? Right now, there is only analysis on the code search task and there is not enough discussion about AST approaches.

- The improvement of GraphCodeBERT over previous SOTA, especially CodeBERT, is marginal. There is no significance test in the results. On Code Clone Detection, the improvement is 0.6% F1. On code translation, the improvement is less than 1% for accuracy. The most improvement is on 5.1 Natural Language Code Search, but the data flow graph and the two newly introduced objectives are not relevant to natural language comment.

- I don't think BLEU is a meaningful metric for code translation and code refinement, and it can be misleading. For example, in Table 4, BLEU is 90+ while the accuracy is less than 10%. The author can consider not reporting BLEU.

Questions
-  Can you please give more details about how to use AST Preorder Traversal and AST Subtree Masking for code search? I am wondering if this is a good usage of AST on this task. Thanks!

Minor
- Notation W is used to denote both parameters in self-attention (Eq (1)) and the code comment. Consider using two different notations.

---

> ### Author Response · Authors · 2020-11-16
> **Respond to AnonReviewer4**
>
> ### Respond to weakness:
> 1.	AST contains attribute information of code tokens like function name and operator, dependency information between code tokens like data flow and so on. In this paper, we mainly leverage dependency relation between variables to demonstrate that semantic-level code structure is useful for pre-training. It is promising to consider function name, condition control and operator as well. We leave them in the future works.
>
> 2. We add t-test on four tasks in the latest draft. For code search, the results show that the improvements of GraphCodeBERT are significant with $p$<0.01 on 5 programming languages. For code translation and other two tasks, GraphCodeBERT significantly outperforms other models with $p$<0.05 and $p$<0.01, respectively. Please kindly find the t-test results on Table 1 to 4 from the updated draft.
>
>    Although data flow and the two newly introduced objectives are not directly relevant to natural language comment, data flow improves the code understanding ability of the model, which helps GraphCodeBERT do better semantic matching between source code and queries, as shown in Figure 4.
>
> 3. Thanks for your suggestion about the BLEU metric. We will consider not reporting BLEU in the final version.
>
> ### Respond to  questions:
> 1.	AST Pre-order Traversal regards AST as a sequence $S$ by linearizing all AST nodes using pre-order traversal algorithm.  We then concatenate the comment $W$ and linearized AST $S$ as the sequence input $X=([CLS], W, [SEP], S, [SEP])$ and add an extra vocabulary for non-terminal nodes. We take $X$ as input to pre-train the model using MLM objective and fine-tune the model on downstream task.
>
> 2. AST Subtree Masking regards AST as a tree. The set of non-terminal symbols of AST is denoted as $T$.  We then concatenate the comment $W$, source code $C$, and the set of non-terminal symbols $T$ as the sequence input $X=([CLS], W, [SEP], C, [SEP], T, [SEP])$ and add an extra vocabulary for non-terminal nodes. Following [1], we introduce tree-based embeddings and a subtree masking function. Tree-based embeddings indicate the position of non-terminal symbols of $T$ in AST.  Subtree masking function is to allow each non-terminal node to attend to only its own subtree descendants in Transformer. Applying tree-based embeddings and subtree masking function could introduce tree structure information of AST into Transformer. We take the sequence $X$ as the input to pre-train the model using MLM objective and then fine-tune on downstream tasks.
>
>    [1] Xuan-Phi Nguyen, Shafiq Joty, Steven Hoi, and Richard Socher. Tree-structured attention with hierarchical accumulation. International Conference on Learning Representations, 2020.
>
> ### Respond to minor:
> Thanks for your suggestion and we will consider using two different notations to represent parameters in self-attention (Eq (1)) and the code comment in the final version.

---

### Official Review · AnonReviewer2 · 2020-10-27
**Simple extension over CodeBERT with small performance improvement for a series of tasks**

**Rating:** 7
**Confidence:** 5

**Review:**

This paper extends CodeBERT to include elements of dataflow in
addition to the comments and sequence of code tokens. Also, the
pretraining includes two tasks that are structure-aware: predicting
dataflow edges and aligning dataflow nodes to code. The paper is well
written, well motivated and the empirical evaluation is quite
thorough. The performance compared to CodeBERT is around 1-2%
additional accuracy for a series of diverse tasks.

Detailed comments:
- In general the paper is well written. There were a few parts that
  required further clarification for me. For example, it is not clear
  how the variable names are represented textually (e.g. x1 - is it
  really x1 or a different way of expressing this in the sequence of
  tokens that is sent to the model). The architecture seems to me to
  be pretty identical to BERT with the exception of the attention
  matrix for the dataflow portion. Is this matrix kept constant
  throughout training? Not clear to me if the dataflow graph is a
  traditional dataflow (as in the one used in compiler analysis). It
  appears to me that it's a slightly simplified version based on AST
  and some idea of data flow. I would contrast and compare with
  "traditional" dataflow graphs (that use SSA - single static
  assignment form - for example). Not exactly clear what you mean by
  "neat". I would find a different word. Neat sounds like "pretty" in
  this context; I don't find it scientific. Do you mean less complex?
  The Noe alignment task was not clear to me at all. Perhaps an
  example would be helpful showing the actual task and what is
  provided as input (I'm guessing the same thing as usual) and how the
  prediction looks like.

- I'm not sure there is a discussion on the experimental methodology
  and hyperparameter tuning. Are the results presented some average
  over several runs? If they are single runs, the difference in
  accuracy for most task is within noise, I would think. Also, I
  particularly find the third decimal a bit much for "stretching"
  result improvements. I think the idea is simple, it gives some
  limited performance improvement over CodeBERT but the model
  architecture is overall similar so it is probably worth it

---

> ### Author Response · Authors · 2020-11-16
> **Respond to AnonReviewer2**
>
> ###   Respond to comments:
>
> 1. We use the byte-pair encoding (BPE) method [1] to tokenize variable names, e.g. “x1” and “max_value” will be tokenized to [‘Ġx’,’1’] and [‘Ġmax’, ‘_’, ‘value’] where ‘Ġ’ is the special token to represent the beginning sub-token of variable names.
>
>    [1] Rico Sennrich, Barry Haddow, and Alexandra Brich. 2016. Neural machine translation of rare words with subword units. In Proceedings of the 54th Annual Meeting of the Association for Computational Linguistic.
>
> 2.	The attention matrix $\rm{softmax}(\frac{Q_iK_i^T}{\sqrt{d_k}}+M)$ is learnable regarding parameters $Q_i$, $K_i$ in the equation 1. $M$ is a pre-defined matrix and keeps 0 or $-\infty$ throughout training to mask attention between unconnected node pairs.
>
> 3.	Dataflow considered in this work ***represents dependency relation between variables***, which is different from the traditional dataflow that covers more semantic information in compiler analysis. We will further clarify the difference in the paper.
>
> 4.	Yes, we mean less complex. Thanks for your suggestion. We will use a more suitable word like “less complex”.
>
> 5.	Thanks for your suggestion, we have added an example with more detailed description for the node alignment task. Please kindly find the Figure 3 and detailed description in the latest draft.
>
> 6. We discussed experiment settings and hyperparameter in Appendix A to D.
>
>    For code clone detection, code translation, and code refinement tasks, the results presented in the paper is to average over five runs using different random seeds. For code search task, we use the same pipeline released by CodeBERT and keep the same random seed for fair comparison.
>
>    In the rebuttal phase, we further conduct t-test for four downstream tasks. For code search, the results show that the improvements of GraphCodeBERT are significant with $p$<0.01 on 5 programming languages. For code translation and other two tasks, GraphCodeBERT significantly outperforms other models with $p$<0.05 and $p$<0.01, respectively. Please kindly find the t-test results on Table 1 to 4 of the updated draft.

---

### Official Review · AnonReviewer3 · 2020-10-28
**Review of Graph Code BERT**

**Rating:** 7
**Confidence:** 3

**Review:**


Overview:
The authors present Graph Code BERT, the first language model that leverages data flow to learn code representation. They use three objective functions: Masked Language Modeling, Edge Prediction, and Node Alignment. They claim their structure-aware pre-training can help improving performance on code-related downstream tasks, including code search, clone detection, code translation, and code refinement.


Reasons to accept:
* The paper is well-written and easy-to-follow.

* It is the first pre-trained model that leverages the data flow structure of code to learn code representation.

* They propose two pre-training tasks for learning representation from source code and data flow. Edge Prediction is more interesting in my opinion.

* They show improvement on four downstream tasks, even though some of them are marginal in my opinion compared to CodeBERT.

Reasons to reject:
* This work is not the first work to conduct language model pretraining for code understanding applications. I will expect more comprehensive comparisons with related work. For example, Svyatkovskiy et al. (2020) propose GPT-C, which is actually a transformer architecture that can also use for both generation and classification tasks. Missing C-BERT (Buratti et al.) as well.

* The 2.4M functions used for "pre-training" is not very persuasive. I think it is more reasonable to crawl existing code repositories for Pre-training. I do not see a particular reason because pre-training without natural language descriptions still makes sense.

Questions & Suggestions:
* Do you find natural language descriptions useful in your experiments? Why if we discard it from pre-training?

* What is the ratio of edge connection? 10%? 20%? The reason I am asking this is that when you do the randomly sampling 20% of nodes Vs in the data flow, how many node pairs you used to calculate the loss? Is the positive-negative ratio balanced? Please give more details about the training objectives.

* Please give more prediction examples or qualitative analysis for each downstream tasks in the Appendix.

---

> ### Author Response · Authors · 2020-11-16
> **Respond to AnonReviewer3**
>
> ### Respond to reject reasons:
> 1.	We did not compare to GPT-C or C-BERT because they are not publicly available. We will contact authors of related works to conduct fair and comprehensive comparisons.
> 2.	Since CodeBERT is pre-trained on 2.4M functions corpus, for fair comparison, we use the same pre-training dataset (2.4M functions) for GraphCodeBERT, which is mentioned in Appendix A.
>
> ### Respond to questions & suggestions:
> 1.	The motivation of utilizing natural language descriptions for pre-training is to support more code-related tasks involving natural language such as code search.  In the experiments of CodeBERT paper, a model pre-trained on only code without using natural language descriptions performs worse than a model pre-trained on text-code pairs. We retain their setting in this work.
>
> 2.	The average number of nodes and the ratio of edge connection are 27 and 8%, respectively. For each example, about 5 nodes will be sampled and 135 node pairs will be used to calculate the loss.
>
> 3.	We have added case study and error analysis for each downstream task. Please kindly find them in Appendix F and G of the latest draft.

---

### Official Review · AnonReviewer1 · 2020-11-02
**Solid work on introducing structure aware tasks and data flow representation to pre-training on code**

**Rating:** 7
**Confidence:** 3

**Review:**

This work address the pretraining over code and text. It proposes to leverage data flow as additional inputs, and add two structure aware pre-training tasks besides the masked token prediction task. The pretrained model is evaluated on four different tasks and outperforms the CodeBERT baselines as well as other pretrained models. Further analysis confirmed the benefits from the additional tasks and data flow input.

Strength:

1. The use of data flow as additional input and the proposal of structure aware tasks are well motivated adaptations of pre-training to code.

2. The extensive experiments and comparisons supported the claim and additional ablation studies and analysis confirmed the benefits from the additional structure-aware tasks and the data flow representation.

3. The comparison with syntactic structures (AST) is interesting and demonstrated the higher potential of semantic structures like data flow. This could be beneficial to other code related tasks.

4. The paper is well written and easy to follow.

Weakness:

1. Since the proposed approach is adding extra inputs, and given the O(n^2) complexity of Transformer, it would help to add some information regarding the increase in compute cost, for example, additional flops or latency due to additional data flow inputs.

Related to the point above, although linearizing the data flow into sequence seems the simplest approach, it might be computationally inefficient given that the data flow graph is probably sparse, thus large number of attention position needs to masked. But maybe the data flow input is much smaller than the code input so that it is not adding too much overhead.

It would also help to add more statistics about the inputs, for example, the average length of the code, the length of the linearized data flow, the length of the linearized AST, etc.

2. Not so much a weakness, but it would help to show some examples, in appendix if the space is limited, where GraphCodeBERT improves over CodeBERT or other models without semantic structures, to give some more intuition.

---

> ### Author Response · Authors · 2020-11-16
> **Respond to AnonReviewer1**
>
> ### Respond to weakness:
>
> 1. By truncating the input sequence, we compared GraphCodeBERT with other Transformer models in the setting of same computing cost. From Figure 4, we can see that GraphCodeBERT performs better than other models when we keep the same length of input sequence e.g. 128. The results demonstrate that adding data flow inputs still benefits although truncating the input sequence.
>
>    In the rebuttal phase, we conduct statistical analysis on the input, including the average length of the code, the length of the linearized data flow, the average length of linearized AST. All numbers are calculated on the CodeSearchNet dataset after tokenization. The statistics show that the average length of linearized data flow is about 14%~22% of code input and three times less than AST. We will add these statistics in the final version. Thanks for your suggestion.
>
>    |                                        | Ruby | Python | Java | Php  | Go   | Javascript |
>    | -------------------------------------- | :--- | :----- | :--- | :--- | :--- | :--------- |
>    | Average length of code                 | 116  | 156    | 131  | 152  | 120  | 149        |
>    | Average length of linearized data flow | 22   | 35     | 25   | 21   | 18   | 32         |
>    | Average length of linearized AST       | 64   | 104    | 75   | 94   | 76   | 94         |
>
> 2. We have added case study and error analysis for each downstream task. Please kindly find them in Appendix F and G of the latest draft.

---

### Author Response · Authors · 2020-11-16
**General Response: Summary of Updated Version**

Thank all reviewers for your suggestions and comments. We have submitted an updated version of the paper based on these comments and included additional experiments.

1 ) We have added an example with more detailed description for node alignment task in Figure 3 suggested by Reviewer 2.

2 ) We have conducted significance test on four downstream tasks in Table 1 to 4 according to Reviewer 2's and Reviewer 4's comments.

3 ) We have added case study and error analysis for each downstream tasks in Appendix F and G suggested by Reviewer 1 and Reviewer 3.

---

### Decision · Program_Chairs · 2021-01-07
**Final Decision**

**Decision:**

Accept (Poster)

**Comment:**

This paper proposes a simple extension to BERT-like pre-training for source code models, which allows incorporation of data flow information. This is a new way of incorporating code structural information into models, and it appears practical and effective. Reviewers are all in favor of accepting the paper.